# OpenReview forum: "Less is Enough: Synthesizing Diverse Data in LLM Feature Space with Sparse Autoencoders"
_ICML.cc/2026/Conference — ICML 2026 spotlight_

### Official Review · Reviewer_NVme · 2026-02-14

**Soundness:** 3
**Presentation:** 3
**Significance:** 3
**Originality:** 3
**Overall Recommendation:** 5
**Confidence:** 4

**Summary:**

This paper focuses the problem of constructing diverse post-training datasets for LLMs by proposing Feature Activation Coverage (FAC), a diversity metric defined in the interpretable feature space of Sparse Autoencoders (SAEs). Building on FAC, they propose FAC Synthesis, a two-stage data synthesis framework that firstly identifies missing SAE features not covered by a seed dataset, and then generates targeted synthetic samples that activate those features using contrastive pair-guided prompting. Experiments span four tasks across three model families demonstrates the outperformance of proposed method.

**Compliance With Llm Reviewing Policy:**

Affirmed.

**Final Justification:**

Upon reviewing the submitted paper and the rebuttal response, I found the work to be of high quality. During the rebuttal period, the authors have comprehensively addressed all the concerns raised. All of the reviewers have provided detailed concerns, and I believe that including such clarifications in the rebuttal would significantly enhance the paper’s quality. I am pleased to support the acceptance of the paper.

**Key Questions For Authors:**

1. Have you experimented with continuous FAC variants? How do they compare in correlation with downstream performance?
2. Any preliminary results on other benchmarks such as  reasoning-heavy benchmark (GSM8K, MATH, code generation)? Even negative results would clarify the method's scope.

**Limitations:**

yes

**Strengths And Weaknesses:**

## Strength 1
The problem is well-motivated and the solution is principled. The gap between text-level diversity metrics and downstream performance is a real issue, and measuring diversity in a model-aware, interpretable feature space is a clean conceptual contribution. The theoretical framework assembles standard tools thoughtfully to justify why feature coverage matters.
## Strength 2
The empirical results are strong and data efficiency is impressive. FAC Synthesis consistently outperforms seven baselines across all four tasks, often by large margins. Matching MAGPIE's AlpacaEval performance with 150× fewer samples is practically significant.
## Weakness 1
The FAC metric binarizes continuous SAE activations via threshold , discarding information about activation strength and frequency. A feature barely activated once is treated identically to one strongly activated across many samples. While parameter sensitivity is studied, the paper does not explore whether a soft coverage metric accounting for activation magnitudes could yield stronger performance correlations or better synthesis guidance.
## Weakness 2
The theoretical analysis has notable gaps between theory and practice. For example, the residual term is acknowledged as potentially large and "not directly optimized." The binary coverage surrogate is quite loose. The connection between contrastive pair construction and conditional entropy reduction is more qualitative intuition than a tight bound.
## Weakness 3
SAE training sensitivity is underexplored. The SAE is trained on specific corpora at a specific layer with a specific architecture. The paper does not study how performance varies with SAE quality

---

> ### Author Rebuttal · Authors · 2026-03-30
>
> We thank the reviewer for the constructive comments, which we address as follows.
>
> ### W1. The FAC metric binarizes continuous SAE activations via threshold, discarding information about activation strength and frequency.
> > Thanks for the suggestion. Activation frequency and magnitude are more important when most features are already covered and the goal shifts to refining their distribution. In our setting, however, the primary bottleneck is that many task-relevant features are entirely absent from the seed dataset. We compare binary FAC against two continuous variants to verify this:
> >
>  >  - **FAC_freq**: weights each feature by the fraction of samples that activate it.
> >   - **FAC_magnitude**: weights each feature by its average activation strength.
> >
> >   | FAC Variant | FAC (%) | AUPRC (%) | vs Baseline |
> >   |---|---|---|---|
> >   | Binary FAC (ours) | 81.78 | 62.60 | +23.63 |
> >   | FAC_freq | 79.44 | 61.74 | +22.77 |
> >   | FAC_magnitude | 83.65 | 63.01 | +24.04 |
> >
> >  FAC_freq performs slightly worse (61.74% vs. 62.60%), because our anchor set (~190K instances) is too small for reliable frequency estimation. FAC_magnitude performs slightly better (63.01%), consistent with Figure 6 (Section 7.6), where higher activation thresholds select more reliably activated features. All three variants stay within a narrow range (61.74-63.01%), confirming that *which* features are covered matters more than how they are weighted. We expect frequency-based weighting to become more useful with larger anchor sets.
>
> ### W2. The theoretical analysis has notable gaps between theory and practice.
> > We appreciate this detailed feedback. We address each point below.
>  (1) **Residual term**: The reviewer is right that $\epsilon_{cond}$ in equation (2) is not directly optimized. However, it is not ignored either. Appendix C analyzes it in detail and introduces retrieval-conditioned prompting as a practical mechanism to upper-bound and reduce it.
> (2) **Binary surrogate**: We agree the binary surrogate is a simplification. However, our FAC variants experiment (W1 above) empirically validates that this simplification does not hurt: binary, frequency-weighted, and magnitude-weighted FAC all yield comparable AUPRC (61.74-63.01%). This suggests the binary approximation captures the dominant signal.
> (3) **Contrastive pairs**: The connection is more than qualitative intuition. Appendix F provides a formal analysis showing that enforcing more missing features monotonically decreases conditional entropy. We will continue to strengthen the theory in future work.
>
> ### W3. SAE training sensitivity is underexplored.
> > Great question! We in fact have studied the effect of different layers in Appendix (Figure 8 and Section K.2). To address the concerm, we further ablate the dictionary size and sparsity (TopK).
> >
> >   **Dictionary Size (TopK = 20 fixed)**
> >
> >   | Dict Size | TopK | Activated Features (L0) | Recon. Loss (L2) | AUPRC (%) |
> >   |---|---|---|---|---|
> >   | 2^14 (16,384) | 20 | 19.75 | 0.1687 | 59.74 |
> >   | 2^16 (65,536) | 20 | 19.41 | 0.1591 | 62.60 |
> >   | 2^17 (131,072) | 20 | 19.21 | 0.1553 | 62.67 |
> >
> >   **TopK Sparsity (Dict size = 2^16 fixed)**
> >
> >   | Dict Size | TopK | Activated Features (L0) | Recon. Loss (L2) | AUPRC (%) |
> >   |---|---|---|---|---|
> >   | 2^16 (65,536) | 10 | 9.8 | 0.1819 | 61.62 |
> >   | 2^16 (65,536) | 20 | 19.41 | 0.1591 | 62.60 |
> >   | 2^16 (65,536) | 40 | 37.68 | 0.1378 | 62.89 |
> >
> >   AUPRC remains above 59% across all configurations, confirming robust performance across SAE hyperparameter choices.
>
> ### Q1. Have you experimented with continuous FAC variants? How do they compare in correlation with downstream performance?
> > Yes, as we reported in comments to W1, binary FAC already achieves Pearson r = 0.95 with downstream AUPRC (Figure 9). The continuous variants (FAC_freq, FAC_magnitude) yield comparable AUPRC (61.74-63.01%), confirming that the binary formulation is already highly effective.
> ### Q2. Any preliminary results on other benchmarks such as reasoning-heavy benchmark (GSM8K, MATH, code generation)? Even negative results would clarify the method's scope.
>
> > Great suggestion! We conducted some preliminary experiments on GSM8K and LiveCodeBench:
> >
> >   | Model | GSM8K-Accuracy (%) | | LiveCodeBench (2025.02-2025.05) pass@1 (%) | |
> >   |---|---|---|---|---|
> >   | | Before fine-tuned | After fine-tuned | Before fine-tuned | After fine-tuned |
> >   | LLaMA-3.1-8B-Instruct | 83.78 | 84.53 | 12.21 | 12.21 |
> >   | Mistral-7B-Instruct | 60.73 | 62.24 | 6.87 | 8.40 |
> >   | Qwen2-7B-Instruct | 85.29 | 85.60 | 10.69 | 11.45 |
> >
> >   The gains are smaller than on our main tasks (up to +1.51% on GSM8K, +1.53% on LiveCodeBench). As discussed in Section 8, reasoning features often emerge from distributed circuits spanning multiple SAE layers, which a single-layer SAE cannot fully capture. Extending to multi-layer SAE is a key direction for **future work**.

---

> > ### Author Rebuttal · Reviewer_NVme · 2026-04-01
> >
> > The authors have comprehensively addressed all the concerns raised. All of the reviewers have provided detailed concerns, and I believe that including such clarifications in the rebuttal would significantly enhance the paper’s quality. I am pleased to support the acceptance of the paper.

---

> > > ### Author Response · Authors · 2026-04-01
> > >
> > > Thank you for your kind support. We are grateful that our rebuttal was able to resolve your concerns. We will incorporate the relevant clarifications into the final version.

---

### Official Review · Reviewer_VCH8 · 2026-03-10

**Soundness:** 3
**Presentation:** 2
**Significance:** 3
**Originality:** 3
**Overall Recommendation:** 5
**Confidence:** 3

**Summary:**

This paper proposes **Feature Activation Coverage (FAC)**, a model-aware diversity metric defined in the sparse autoencoder (SAE) feature space of LLMs, and a synthesis framework (FAC Synthesis) that targets missing task-relevant features to generate synthetic data. The authors provide a theoretical framing that links downstream generalization to distributional alignment in feature space and sampling error controlled by dataset uncertainty, and implement a two-step, contrastive prompting strategy to reliably activate target features. Experiments across four tasks and three model families show empirical gains, a high correlation between FAC and downstream metrics, and encouraging cross-model transfer of SAE-identified features.

**Compliance With Llm Reviewing Policy:**

Affirmed.

**Final Justification:**

All my concerns have been solved. Taking into account the overall quality and contributions of this work, I maintain my judgment of Accept.

**Key Questions For Authors:**

- As mentioned in weaknesses, does the author believe that incorporating activation frequency and feature importance is necessary for a robust evaluation? Moreover, the calculation of FAC relies heavily on task-relevant features generated by an LLM (gpt-4o-mini). How do the authors measure the quality and coverage of these generated features?

- In Section 5, the author states that traditional methods (lexical similarity, heuristic rules, reward models) are sensitive to linguistic variations irrelevant to the target task. While intuitive, this is a strong claim. Is there empirical evidence or prior literature cited in the paper to support this as a field-wide consensus? Furthermore, it is not immediately clear why SAE was chosen as the primary encoding mechanism over more conventional dense embeddings. Could the authors provide a comparative analysis or a theoretical justification?

- The proposed method essentially measures the similarity between the generated dataset $D_{gen}$ and the target domain $D$ via SAE encoding. Is the observed performance gain primarily a result of the SAE’s encoding accuracy, or the FAC metric's ability to measure similarity in the $\Delta TV(D, D_{gen})$ space? Providing an ablation study could be more reliable.

**Limitations:**

yes

**Strengths And Weaknesses:**

**Strengths:**
- The work introduces an innovative perspective by utilizing SAE and FAC to quantify data diversity, providing an interpretable measure of task-relevant features on synthetic data.
- The method outperforms SOTA by utilizing only 2,000 synthetic samples, achieving a 150× reduction in data volume compared to baselines like MAGPIE.
- This paper validates the existence of a shared feature space across diverse model families, demonstrating cross-model knowledge transfer is possible.

**Weaknesses:**
- There are some gaps between theory and implementation. Equation (2) offers a general bound based on a latent representation space $\mathcal{Z}$. However, while the author suggests methods to control the conditioning error $\varepsilon_{cond}$ in Appendix, it lacks a mathematical proof or empirical bound demonstrating that SAE provide superior convergence rates or tighter $\varepsilon_{cond}$ control compared to standard latent representations (e.g., traditional dense Embeddings). Consequently, the FAC metric appears to be a heuristic design choice inspired by the SAE architecture rather than a direct analytical result derived from the theoretical upper bound.
- The proposed FAC metric is limited. As a binary, uniformly weighted indicator, it fails to incorporate essential distributional information such as activation frequency and feature importance. The metric does not distinguish between a synthetic dataset where a feature appears in 1% of samples versus 50%, which is crucial for matching the true distribution $P_Z$. And all task-relevant features are weighted equally in Equation (5), ignoring the fact that some features may be more critical for downstream performance than others.
- Some instances are incomplete formatting. For example,  in Section 2, there is a wrong citation: "We provide a full discussion of related work in Appendix ??." Please verify all LaTeX labels and ensure that the appendix and supplementary materials are properly linked.

---

> ### Author Rebuttal · Authors · 2026-03-30
>
> **W1.** Thank you for raising this point. The bound in this paper is intentionally general (building on Zheng et al. [1]). It decomposes generalization error into distribution divergence and sampling error. We adopt a simple but effective approach to address both: FAC reduces distribution divergence (Section 5), and two-step contrastive synthesis reduces sampling error (Section 6). We chose SAE because each feature maps to one semantic concept and can be directly used in a data synthesis prompt. Our ablation further compares against the suggested embedding-based alternative by reviewer, implemented by replacing SAE features with dense embeddings and defining coverage over embedding clusters; empirically, this variant performs worse than our SAE-based FAC.
>
> | Method | Feature Space | AUPRC (%) | vs Baseline |
> |---|---|---|---|
> | Baseline | — | 38.97 | — |
>  | Dense Cluster + Coverage | Raw embeddings | 45.57 | +6.6 |
> | SAE + Random | SAE features | 50.61 | +11.64 |
>  | SAE + FAC (Ours) | SAE features | 62.60 | +23.63 |
>
> > [1] Zheng C, Wu G, Li C. Toward understanding generative data augmentation. 2023.
>
> **W2.** We agree this is worth investigating. Activation frequency and feature importance are more critical when the seed dataset already covers most features and the goal is to refine their distribution. However, in our setting, the primary bottleneck is that many task-relevant features are entirely absent from the seed dataset. In this case, whether a feature is expressed is the dominant signal. To verify this empirically, we compare binary FAC against two weighted variants:
> - **FAC_freq**: weights each feature by the fraction of samples in which it is activated.
> - **FAC_magnitude**: weights each feature by its average activation strength across samples.
>
>  | FAC Variant | FAC (%) | AUPRC (%) | vs Baseline |
>  |---|---|---|---|
>  | Binary FAC (ours) | 81.78 | 62.60 | +23.63 |
> | FAC_freq | 79.44 | 61.74 | +22.77 |
>  | FAC_magnitude | 83.65 | 63.01 | +24.04 |
>
>  FAC_freq performs slightly worse (61.74% vs. 62.60%). This is expected: our anchor set (~190K instances) is relatively small, so frequency estimates are noisy and introduce bias rather than useful signal. FAC_magnitude performs slightly better (63.01%), consistent with Figure 6 (Section 7.6), where higher activation thresholds select more reliably activated features. All three variants stay within a narrow range (61.74-63.01%), confirming that *which* features are covered matters more than how they are weighted. We plan to investigate frequency-based weighting with larger anchor sets in future work.
>
> **W3.** Thank you for catching this. We will verify all LaTeX labels and cross-references in the revision.
>
> **Q1.** Thank you for this question. For the first part, our W2 response confirms that binary coverage is sufficient at our current scale. For the second part, Table 5 in the paper reports 85% human agreement with GPT-4o-mini across three tasks. We further test robustness by injecting two types of noise into the annotation pipeline: (1) **Mislabel**: randomly flip 10% or 20% of the task-relevant labels; (2) **Poor summary**: truncate the last 20% of each feature summary, or replace 20% of summaries with those from random features.
>
> | Setting | Noise (%) | Jaccard(F_miss) (%) | FAC (%) | AUPRC (%) |
>  |---|---|---|---|---|
> | Original | 0 | 100.00 | 81.78 | 62.60 |
> | Mislabel | 10 | 82.13 | 77.10 | 61.55 |
> | Mislabel | 20 | 66.54 | 72.90 | 59.69 |
> | Poor summary (truncation) | 20 | 100.00 | 78.97 | 61.75 |
> | Poor summary (mismatch) | 20 | 100.00 | 70.56 | 58.85 |
>
> Even with 20% annotation noise, AUPRC drops by at most ~4%, yet still exceeds all baselines in table 1 of the paper.
>
> **Q2.**
> Thanks for the question. Yes to both. Jung et al. (2025) [2] show that n-grams and embedding-based diversity signals cannot reliably predict target-task performance. Guo et al. (2025) [3] show that lexical metrics cannot adequately capture semantic diversity. Our own Figure 9 (Appendix K.5) confirms: embedding-level metrics correlate weakly with AUPRC (r <= 0.37), while FAC achieves r = 0.95.
>
> As shown in W1, k-means over dense embeddings gives only +6.6%, compared to +23.63% for SAE+FAC, because SAE features are directly decomposable into promptable concepts.
>
> > [2] Jung J, Han S, Lu X, et al. Prismatic synthesis: Gradient-based data diversification boosts generalization in llm reasoning. 2025. [3] Guo Y, Shang G, Clavel C. Benchmarking linguistic diversity of large language models. 2025.
>
> **Q3.** Thanks for the question. Both components are essential. As shown in the ablation study table in W1 above, SAE + Random (+11.64) already outperforms Dense Cluster + Coverage (+6.6), showing SAE encoding alone is better than dense embeddings. Adding FAC guidance doubles the gain again (+23.63 vs. +11.64), showing FAC is equally important. SAE gives us interpretable features; FAC tells us which ones to cover.

---

> > ### Author Rebuttal · Reviewer_VCH8 · 2026-04-02
> >
> > Thank the author for detailed rebuttal and the additional experiments provided. The newly included ablation studies on the SAE + FAC modules, two FAC variants, and the mislabel experiments have addressed some of my concerns. The experimental results demonstrate the validity of the proposed method.
> >
> > However, while the author validates the effectiveness of the method with empirical results and provides an intuitive justification for incorporating SAEs, there is still a gap between theory and implementation. Specifically, the response does not yet provide a rigorous theoretical derivation or analysis to justify the necessity of SAEs over traditional dense embeddings. To enhance the rigor of the theoretical component, the authors could consider demonstrating that SAEs offer a tighter error bound compared to dense embeddings under the proposed framework.
> >
> > In summary, I will maintain my current score.

---

> > > ### Author Response · Authors · 2026-04-03
> > >
> > > We sincerely appreciate the reviewer for this insightful and important point. Following this suggestion, we have added a formal derivation showing that SAEs provide a tighter coverage surrogate than dense embeddings under our generalization framework. The detailed proof is available at https://anonymous.4open.science/r/Rebuttal-258E/Appendix_sae_vs_dense_proof.pdf.
> > >
> > > Key idea. The paper uses feature coverage as the operational surrogate for reducing the distribution mismatch between anchor and synthetic data. The core question is therefore: which representation (SAE features or clusters derived from dense embeddings) provides a more accurate coverage surrogate for features?
> > >
> > > SAE is exact; dense clustering is not. SAE tracks feature coverage directly, so the coverage indicator is exact by construction. Dense clustering cannot track feature coverage directly and instead relies on cluster coverage as a proxy. Since multiple distinct features may map to the same cluster, an observed cluster does not guarantee that all of its associated features are actually covered. We quantify this mismatch by the dense proxy error, denoted by $L_{merge}$.
> > >
> > > Main result (Proposition 1). The surrogate disagreement gap is lower bounded by dense proxy error $L_{merge}$. In other words, dense clustering can introduce an additional mismatch relative to the paper’s feature coverage definition, whereas SAE does not incur this mismatch because it operates on the coverage indicator directly. The full proof is provided in the appendix linked above.
> > >
> > > Conclusion. Proposition 1 shows that, in our framework, SAE and dense clustering do not track the same coverage object: SAE tracks feature coverage directly, while dense clustering relies on a proxy that can introduce additional mismatch. This is also consistent with our ablation: SAE + Random outperforms Dense Cluster + Coverage (50.61% vs. 45.57% AUPRC). We thank the reviewer for motivating this clarification!

---

### Official Review · Reviewer_4Uic · 2026-03-10

**Soundness:** 3
**Presentation:** 3
**Significance:** 3
**Originality:** 3
**Overall Recommendation:** 5
**Confidence:** 3

**Summary:**

The paper proposes Feature Activation Coverage (FAC) as a new way to measure and improve the diversity of post‑training datasets by looking directly at an LLM’s internal feature space rather than surface‑level text diversity. Building on this insight, the authors introduce FAC Synthesis, a coverage‑guided data generation method that creates synthetic samples specifically designed to activate missing features.
Using Sparse Autoencoders (SAEs), the authors extract interpretable latent features from model activations and determine which task‑relevant features are missing from existing datasets.

Using a two‑step contrastive prompting strategy, the method reliably produces high‑quality, feature‑targeted examples and improves data efficiency: with only 2,000 synthetic samples, it matches the performance of MAGPIE, which requires 300,000. FAC Synthesis outperforms all baselines across toxicity detection, reward modeling, behavior steering, and instruction following, and its SAE‑derived features transfer effectively across model families including LLaMA, Mistral, and Qwen, indicating a shared interpretable feature space.

**Compliance With Llm Reviewing Policy:**

Affirmed.

**Key Questions For Authors:**

1. How exactly are features deemed task‑relevant vs. irrelevant?
2. Can the authors quantify how mistakes in SAE feature interpretation (mislabeling or poor summaries) propagate through Fmiss​, sample synthesis, and final task performance?

**Limitations:**

Yes

**Strengths And Weaknesses:**

Strengths:
1. The authors provide clear theoretical foundations, including a generalisation error upper bound that decomposes performance into distribution gap and sampling error, and they tie this directly to the proposed objective of improving feature coverage.
2. Robust validation with four diverse tasks :toxicity detection, reward modeling, behavior steering, instruction following tested across three model families LLaMA, Mistral, Qwen.
3. Well defined ablation studies on activation thresholds, generator models, feature budgets, and multi‑round self‑improvement.

Weaknesses:
1. The paper does not evaluate how errors in feature interpretation propagate through the pipeline.
2. The paper does not discuss whether activations reflect realistic task semantics or whether the coverage introduces spurious correlations.

---

> ### Author Rebuttal · Authors · 2026-03-30
>
> We thank the reviewer for the detailed comments, which we address as follows.
>
> ### W1. The paper does not evaluate how errors in feature interpretation propagate through the pipeline.
> > Thanks for the suggestion! We have conducted a robustness study that injects noise into the annotation pipeline under five settings:
> >
> >   - **Original**: no perturbation, using the default GPT-4o-mini annotations.
> >   - **Mislabel (10%/20%)**: randomly flip 10% or 20% of the task-relevant labels.
> >   - **Poor summary (truncation)**: truncate the last 20% of the feature summaries.
> >   - **Poor summary (mismatch)**: for 20% of the features, replace their summaries with summaries from other randomly selected features.
> >
> >  Mislabeling affects F_miss identification (Jaccard drops to 66.54% at 20% noise), which reduces FAC and AUPRC. Poor summaries do not change F_miss (Jaccard stays 100%) because the labels are correct, but they degrade synthesis quality. In all cases, AUPRC drops by at most ~4%, yet still exceeds all baselines in table 1 of the paper.
> >
> >   | Setting | Noise (%) | Jaccard(F_miss) (%) | FAC (%) | AUPRC (%) |
> >   |---|---|---|---|---|
> >   | Original | 0 | 100.00 | 81.78 | 62.60 |
> >   | Mislabel | 10 | 82.13 | 77.10 | 61.55 |
> >   | Mislabel | 20 | 66.54 | 72.90 | 59.69 |
> >   | Poor summary (truncation) | 20 | 100.00 | 78.97 | 61.75 |
> >   | Poor summary (mismatch) | 20 | 100.00 | 70.56 | 58.85 |
>
> ### W2. The paper does not discuss whether activations reflect realistic task semantics or whether the coverage introduces spurious correlations.
> >   While we appreciate the concern, we would like to remind the reviewer that we have in fact conducted experiments to confirm that activations reflect realistic task semantics in alignment with humans. Table 5 in the paper shows 85% human agreement with GPT-4o-mini annotations across three tasks, confirming that the identified features match human judgment. The robustness study in W1 above further shows that even with 20% noise, AUPRC drops by at most ~4%.
>
> ### Q1. How exactly are features deemed task‑relevant vs. irrelevant?
> >   Thank you for the question. We use a three-step process: (1) retrieve the top-10 activated text spans for each feature, (2) ask GPT-4o-mini to summarize what they share, and (3) apply a four-level rubric to judge task relevance. Features that pass the judgement are labeled as task-relevant. The full prompts and rubrics are on pages 41-44 (Appendix J).
>
> ### Q2. Can the authors quantify how mistakes in SAE feature interpretation (mislabeling or poor summaries) propagate through Fmiss​, sample synthesis, and final task performance?
> >   Thank you for this suggestion. As we reported in our responses to W1, mislabeling corrupts relevance labels and affects F_miss (Jaccard drops to 66.54% at 20% noise). Poor summaries corrupt the synthesis prompts but not F_miss. In both cases, AUPRC stays within ~4% of the original.

---

> > ### Author Rebuttal · Reviewer_4Uic · 2026-04-01
> >
> > I thank the authors for thoroughly addressing all the concerns raised. I am pleased to support acceptance of the paper, and strongly recommend the inclusion of the improvements and clarifications outlined in the authors’ rebuttal responses to all the reviewers.

---

> > > ### Author Response · Authors · 2026-04-01
> > >
> > > Thank you for your supportive feedback. We are glad that our rebuttal addressed your concerns, and we appreciate your recommendation to include these clarifications in the final version.

---

### Official Review · Reviewer_hVmE · 2026-03-14

**Soundness:** 2
**Presentation:** 2
**Significance:** 2
**Originality:** 3
**Overall Recommendation:** 3
**Confidence:** 4

**Summary:**

This paper proposes Feature Activation Coverage (FAC), a diversity metric defined over SAE features rather than text space, and a FAC Synthesis pipeline that identifies task-relevant missing features from a seed dataset and generates synthetic examples to cover them. The method uses an SAE to map examples into a sparse feature space, defines FAC as the fraction of task-relevant anchor features covered by the generated dataset, and then synthesizes new data targeting the missing-feature set. The paper also adds a two-step contrastive generation procedure intended to make target-feature activation more reliable.

**Compliance With Llm Reviewing Policy:**

Affirmed.

**Final Justification:**

See rebuttal comment.

**Key Questions For Authors:**

My questions are directly associated to the concerns listed above, so please answer those.

**Limitations:**

No, most of the concerns listed are not directly addressed.

**Strengths And Weaknesses:**

Overall, the empirical results are promising. The method outperforms the listed baselines across toxicity detection, reward modeling, behavior steering, and instruction following, and the paper reports strong correlation between FAC and downstream performance. The authors also show that the two-step generation strategy yields higher FAC than one-step synthesis, and they present some evidence of cross-model transfer of the discovered features.

A major strength is that the paper has a clear and interesting core idea: measuring synthetic-data diversity in model feature space rather than text space. That is a meaningful shift from generic lexical or embedding diversity metrics, and the empirical results suggest the idea may be useful in practice. The reported gains across four tasks are strong, and the data-efficiency claim relative to MAGPIE is interesting.

My concerns are:
1. It is unclear how much of the gain actually comes from the SAE representation itself. The paper compares FAC against standard baselines and compares two-step vs. one-step synthesis, but I did not see an ablation against simpler alternatives such as using raw LLM hidden states, embeddings, clustering, or feature spaces derived without an SAE. As written, the paper argues that SAEs are interpretable and useful, but does not really isolate whether they are necessary for the gains.

2. FAC seems incomplete as a synthesis objective once coverage saturates. FAC is defined as set coverage of task-relevant features found in the anchor set, so once each such feature has been activated at least once, the metric no longer says what additional data would be valuable. That makes FAC feel more like a recall-style objective than a full characterization of data quality: it does not distinguish one narrow example per feature from broad coverage of modes, contexts, or feature combinations.

3. A third weakness is the role of the anchor set and auxiliary resources. The paper states that the anchor set is built from large-scale instruction-preference corpora and that the SAE is pretrained on about 711,000 unique queries. The authors also note that baseline synthetic sample budgets are matched, which is good, but the overall method still benefits from substantial extra data and compute outside the final synthetic training budget. This matters for how to interpret the efficiency claim.

4. Relatedly, I think the paper is missing an especially important baseline: using the anchor data directly. Since the method assumes access to a sizable anchor set and uses it to define the target feature coverage, it would be very helpful to compare against stronger non-synthesis alternatives such as direct fine-tuning on the anchor set, feature-guided subset selection from the anchor set, or retrieval/reweighting of anchor examples. Right now it is hard to tell whether the gain comes from synthesis per se, or from having a good feature-targeting signal. In fact, using the anchor data directly should get 100% FAC (and therefore be the optimal thing to do according to the authors).

5. Another weakness is that the claimed task-specificity feels looser than advertised. The SAE itself is trained unsupervised on broad instruction-tuning data; the task-specific part seems to come mainly from the later step where GPT-4o-mini and rubric-based annotation identify which SAE features are task-relevant. That can still be useful, but it means the representation is not inherently task-specific.

6. I also found the theory-to-method connection fairly loose. The paper motivates the approach using a generic generalization bound with TV distance, then upper-bounds TV by KL in feature space, but the practical method ultimately optimizes a thresholded binary coverage heuristic. Likewise, the PAC-Bayesian / entropy discussion motivates the two-step generation design, but the connection to the actual algorithm feels more suggestive than tight. The theory is helpful as high-level motivation, but currently reads as more decorative than actionable or helpful.

7. Finally, some important implementation details are present but somewhat buried. The appendix explains that feature semantics come from the top-10 activated spans and are summarized and verified using GPT-4o-mini, with human auditing of selected features. That is useful, but it also means the pipeline depends on an additional LLM-based interpretation layer whose reproducibility and robustness deserve more discussion in the main paper.

---

> ### Author Rebuttal · Authors · 2026-03-30
>
> **W1.** SAE is necessary because each SAE feature sparsely maps to one semantic concept, which can be directly used in synthesis prompts for guiding targeted data synthesis. Raw LLM hidden states, dense embeddings, and clusters lack this property, where the synthesis prompts derived from them are less effective. Appendix K.5 (Figure 9) already shows a weak correlation of embedding-level metrics with AUPRC (r <= 0.37). We further verify this with two exps: (1) **Dense Cluster + Coverage** replaces SAE with k-means on raw embeddings; (2) **SAE + Random** randomly samples features from the SAE space. Random SAE selection (+11.64) nearly doubles the cluster-based gain (+6.6), and FAC doubles it again (+23.63).
>
>   | Method | Feature Space | AUPRC (%) | vs Baseline |
>   |---|---|---|---|
>   | Baseline | — | 38.97 | — |
>   | Dense Cluster + Coverage | Raw embeddings | 45.57 | +6.6 |
>   | SAE + Random | SAE features | 50.61 | +11.64 |
>   | SAE + FAC (Ours) | SAE features | 62.60 | +23.63 |
>
> **W2.** We agree that FAC is not meant to fully characterize data quality. Its role is narrower: to identify missing task-relevant features and guide what to synthesize next. This does not make FAC incomplete for our setting, because the effective coverage ceiling is not fixed: additional task-relevant features can be uncovered by expanding the anchor set or by enriching the SAE feature space, e.g., through larger SAE dictionaries or cross-layer feature extraction. Feature combinations (e.g., specific contexts) may further improve results, but may also introduce noise, which we will study in future work.
>
> **W3.** While we understand your concern, our efficiency claim is defined in terms of task-specific data synthesis: Given the same synthesis budget, FAC Synthesis produces better training data than baselines (Table 1, Appendix I). This follows the standard protocol [1,2] which also uses pre-existing resources not counted toward the synthesis budget, to ensure a fair comparison. Also, we emphasize that the publicly available instruction data requires no task-specific labels and is relatively easy to access. The SAE is trained once (Appendix K.2) and reused across all tasks, which makes the overall pipeline data-efficient and scalable.
>
>   > [1] Taori R, et al. Alpaca: A strong, replicable instruction-following model. 2023.
>   > [2] Xu C, et al. Wizardlm: Empowering large pre-trained language models to follow complex instructions. 2025.
>
> **W4.** Thank you for the comment. We have already conducted this comparison in the paper. Table 9 (Toxicity Detection) and Table 10 (Reward Modeling) in Appendix compare FAC Synthesis against direct fine-tuning on the subset of anchor data. FAC Synthesis outperforms in both cases.
>
> **W5.** In Figure 2, “task-specific” does not mean that we train a separate SAE for each task. Rather, it refers to the process of selecting task-relevant features from a shared SAE feature space. The SAE provides a general representation, and the task-specificity comes from the downstream feature selection step, which determines the effective task distribution $P_Z$. Training a separate task-specific SAE would require substantial in-domain data, which is precisely the limitation our method is intended to overcome. In addition, an SAE trained on only limited seed data would be unlikely to expose the rare or long-tail features that our pipeline aims to discover. Empirically, the anchor set constructed in Section 6 provides sufficient coverage to support all four tasks considered in our experiments.
>
> **W6.** We respectfully disagree that the theory is primarily decorative. The bound in equation (1) provides the **principled foundation** for our algorithm. Following Zheng et al (Section 3.1)[3], it decomposes generalization error into two terms: (1) distribution divergence, and (2) sampling error. Our methods are designed to **explicitly target these two terms**: FAC reduces (1), and the two-step contrastive synthesis reduces (2). While the final implementations may look straightforward, these are not ad hoc choices; they serve as tractable proxies for the quantities in the bound. Importantly, our ablation validates that both terms matter: removing FAC drops AUPRC from 62.60% to 50.61%, and removing SAE drops it further to 45.57%. Future directions for tightening the bound are discussed in Appendix C and D.
>
>   > [3] Zheng C, et al. Toward understanding generative data augmentation. 2023.
>
> **W7.** Table 5 in the paper reports 84-86% human agreement with GPT-4o-mini annotations. Additionally, we test robustness by injecting two types of noise: mislabeling (10%/20% of relevance labels flipped) and poor summaries (20% truncated or replaced with random descriptions). Even with 20% noise, AUPRC drops by at most ~4%. Improving the reliability of LLM-based feature annotation is an important direction for future work. We will move the pipeline description into the main paper in revision.

---

> > ### Author Rebuttal · Reviewer_hVmE · 2026-04-03
> >
> > I thank the authors for their response. Broadly, my concerns still hold: the setting is complex (involving many different models trained on different subsets of data, human-written rubrics, etc.), the theory is still in my opinion very loosly tied to the method, and the exposition is unclear. The rebuttal helped to argue for why SAEs are particularly useful (e.g., they can be used to prompt a model to generate data) though even with random features, one could still construct contrastive pairs with different values for those features, so I'm increasing my score slightly.

---

> > > ### Author Response · Authors · 2026-04-04
> > >
> > > We thank the reviewer for the time and effort devoted to carefully understanding and reviewing our paper. We would like to clarify three points that relevant to the remaining concerns.
> > >
> > > First, we want to clarify that our method provides a simple but effective way to measure data diversity and guide data synthesis. As shown in Figure 2, it consists of three main steps: identifying the missing feature set, synthesizing targeted data, and fine-tuning the model, all built on the **one model backbone**. What makes the setting appear broad is not the method itself, but the fact that we evaluate the same pipeline across multiple tasks and model families. We also emphasize that the human-written rubrics used to identify task-relevant features for the four tasks are based on prior published work, as described in Appendix J (“**Identifying Task-relevant Features**”). More broadly, human-written prompts are already standard in many synthetic data baselines, including Alpaca, Evol-Instruct, CoT-Self-Instruct, and SAO in Table 1. The design of better rubrics and prompts is a broader open problem in synthetic data research, rather than an issue specific to our method. Even under this standard setup, Table 1 shows that the proposed method already performs strongly.
> > >
> > > Second, we understand the reviewer’s concern and would like to clarify the role of the theory. Our goal in building the theoretical framework is **threefold**: (1) to explain why SAE and the two-step generation design are well suited to this task; (2) to explain why SAE provides a better control space than dense embeddings, for which we provide a formal derivation in the appendix (https://anonymous.4open.science/r/Rebuttal-258E/Appendix_sae_vs_dense_proof.pdf); and (3) to offer practical guidance for the method itself. For example, Appendix C shows why retrieved spans should be added to the prompt, which supports the retrieval-conditioned prompting step in our method design. Appendix F explains from the perspective of conditional entropy why contrastive data construction is essential in our two-step generation design. Regarding exposition, we would greatly appreciate any specific pointers on which parts were unclear. We will carefully revise those parts in the revision to make the connection between theory and method more explicit.
> > >
> > > Third, we would like to emphasize the **broader value of the proposed framework**. Its key advantage is that it does not only measure data diversity more accurately in the model’s own feature space, but also turns this signal into a direct guide for targeted synthesis. **1.** This makes it potentially useful across **many stages of model training**, including pre-training, supervised fine-tuning, preference alignment, and reinforcement learning, as well as for **multiple tasks** where seed data is limited. **2.** More importantly, the method can even work with only an anchor set: it can automatically identify task-relevant features from anchor data, synthesize data to cover missing features, and naturally support **self-improvement** by re-detecting newly missing features after fine-tuning, which is confirmed in Appendix K.11 of the paper. **3.** In addition, the cross-model transferable features we observe suggest that this framework could reduce the need to rediscover synthesis signals for each new model, making data construction **more scalable** and useful across model families. We believe this framework points to a new direction for data-centric LLM research: identifying the metrics that truly matter for task performance, and using them to understand the real role of data.

---

### Decision · Program_Chairs · 2026-04-30

**Decision:**

Accept (spotlight)

**Comment:**

This paper defines a metric for the diversity of a dataset in its consequent SAE feature activations. Using this diversity metric, the authors propose to generate synthetic data to fill in any missing task-relevant features. This is an intriguing and novel application of current interpretability methods. The paper includes some theoretical results as well as promising empirical results with deep analysis.

The main concerns seem to be the amount of overhead required in order to apply these methods—all of which is specific to a given task setting---but it seems likely that major elements of this pipeline could be efficiently automated eventually.